# Genome-Wide Identification and Expression Analysis of the Aquaporin Gene Family in *Lycium barbarum* during Fruit Ripening and Seedling Response to Heat Stress

Wei He [1,†], Mingyu Liu [1,†], Xiaoya Qin [2], Aihua Liang [3,4], Yan Chen [1], Yue Yin [2], Ken Qin [2,*] and Zixin Mu [1,3,4,*]

1   College of Life Sciences, Northwest A&F University, Yangling, Xianyang 712100, China
2   National Wolfberry Engineering Research Center, Ningxia Academy of Agriculture and Forestry Sciences, Yinchuan 750002, China
3   College of Life Sciences & Technology, Tarim University, Alar 843301, China
4   State Key Laboratory Breeding Base for the Protection and Utilization of Biological Resources in Tarim Basin Co-Funded by Xinjiang Corps and the Ministry of Science and Technology, Aler 843300, China
*   Correspondence: qinken7@163.com (K.Q.); muzx@nwsuaf.edu.cn (Z.M.)
†   These authors contributed equally to this work.

**Abstract:** Plant–water relations mediated by aquaporins (AQPs) play vital roles in both key plant growth processes and responses to environmental challenges. As a well-known medicinal and edible plant, the harsh natural growth habitat endows *Lycium* plants with ideal materials for stress biology research. However, the details of their molecular switch for water transport remain unclear. In the present work, we first identified and characterized AQP family genes from *Lycium* (*L.*) *barbarum* at the genome scale and conducted systemic bioinformatics and expression analyses. The results showed that there were 38 *Lycium barbarum* AQPs (LbAQPs) in *L. barbarum*, which were classified into four subfamilies, including 17 LbPIP, 9 LbTIP, 10 LbNIP, and 2 LbXIP. Their encoded genes were unevenly distributed on all 12 chromosomes, except chromosome 10. Three of these genes encoded truncated proteins and three genes underwent clear gene duplication events. Cis-acting element analysis indicated that the expression of *LbAQPs* may be mainly regulated by biotic/abiotic stress, phytohormones and light. The qRT-PCR assay indicated that this family of genes presented a clear tissue-specific expression pattern, in which most of the genes had maximal transcript levels in roots, stems, and leaves, while there were relatively lower levels in flowers and fruits. Most of the *LbAQP* genes were downregulated during *L. barbarum* fruit ripening and presented a negative correlation with the fruit relative water content (RWC). Most of their transcripts presented a quick and sharp upregulation response to heat stress following exposure of the 2-month-old seedlings to a 42 °C temperature for 0, 1, 3, 12, or 24 h. Our results proposed that *LbAQPs* were involved in *L. barbarum* key development events and abiotic stress responses, which may lay a foundation for further studying the molecular mechanism of the water relationship of *Lycium* plants, especially in harsh environments.

**Keywords:** *Lycium barbarum*; aquaporins; genome-wide; water relations; fruit ripening; heat stress

## 1. Introduction

Plant water transport and related molecular components are responsive to an extremely wide array of environmental and hormonal signals, which is an essential process throughout their life cycle [1]. It is well known that the plant water balance can usually be broken down by two key physiological processes, namely, expansion growth and stress response [2,3]. In both cases, ensuring water transport to the growth centers or avoiding water loss means that maintaining water homoeostasis is vital for whole-plant development and survival [1]. This is especially true under global climate change, in which plants are subjected to incorporated drought and heat stress, which further deteriorates the plant

water status compared with the exertion of single environmental factors. Maintaining plant water balance, therefore, is becoming increasingly indispensable for both sustainable agriculture and ecosystems.

Aquaporins (AQPs), also called water channels or major intrinsic proteins (MIPs), are characterized by six transmembrane domains that together facilitate the transport of water and a variety of low-molecular-weight solutes. The greatest numbers of AQP isoforms are present in plants versus animals and microorganisms, implying their indispensable role in plant sessile living and environmental response [4,5]. The increase in AQP number during the evolution of plants from aquatic to terrestrial and from lower to higher further highlighted the importance of AQPs in the adaptation of higher plants to land life [6,7]. Plant AQPs are present in the plasma and intracellular membranes of most plant cells and play central roles in various physiological processes by ensuring cell-to-cell water transport and, to a lesser extent, single-cell osmotic regulation [8]. These processes include stomatal and leaf movements, seed dormancy and germination, plant growth and development, $CO_2$ fixation, nutrient allocation and toxicity, ROS detoxification and signaling, whole-plant water transport and transpiration, plant reproduction, and abiotic and biotic stress responses [6,9–12]. Through genome sequencing technology, AQP gene families have been comprehensively identified in approximately 50 various plant species, covering algae, mosses, lycophytes, monocots, and dicots [6,11,12].

AQPs are generally classified into five subfamilies in higher plants on the basis of their localizations and amino acid sequences. These include plasma membrane intrinsic proteins (PIPs), tonoplast intrinsic proteins (TIPs), nodulin 26-like intrinsic proteins (NIPs), small basic intrinsic proteins (SIPs), and uncategorized (X) intrinsic proteins (XIPs). The two types of PIPs are PIP1 s and PIP2s, which have differences in amino acid length, amino acid substitution, and water permeability. TIP AQPs exist in more divergent forms than PIPs, in which they are further divided into five isoforms, TIP1, TIP2, TIP3, TIP4, and TIP5, on the basis of their sequence homologies. NIP proteins are also present in divergent forms, as they are divided into five subgroups (NIP1, NIP2, NIP3, NIP4, and NIP). In *Arabidopsis*, NIP isoforms include AtNIP1;1, AtNIP1;2, AtNIP2;1, AtNIP3;1, AtNIP4;1, and AtNIP4;2. XIP is generally found in plasma membranes and is known to be permeable to the largest uncharged solutes, such as urea, boric acid, glycerol, and $H_2O_2$ [6]. In addition, the entire XIP subfamily is absent in both Arabidopsis and rice, presenting its high variation among plant species.

Aquaporins certainly exert a crucial function in alleviating abiotic stress by transporting water and other small molecules to maintain cellular homeostasis [9,10]. Although AQP exists as a large protein family in plants, an increasing number of genetic assays have demonstrated that a single AQP gene can orchestrate the whole plant function, implying its potential application in crop improvement when facing extreme climate challenges [9,13–15]. Taken together, in-depth research on AQP function and regulatory mechanisms will provide a breakthrough point to reveal the molecular mechanism underlying plant water balance. Identification and characterization of AQP family genes in more plant species, including medicinal plants, ex. *Lycium* plants, are, therefore, indispensable both for basic research and human health care practices.

Goji *Lycium* of the Solanaceae family contains ~80 species, representing important food plants and enjoying a reputation as one of the world's most economically and medicinally valuable fruit crops [16–19]. It has been reported that *Lycium* species present a fragmented distribution pattern at high altitudes in the subtropics to temperate regions but are absent in tropical regions. Correspondingly, the temperature requirement for these *Lycium* species is consistent with its geographical distribution—they are sensitive to heat stress and, therefore, significantly reduce yield and fruit quality under high temperature. Referring to a previous study, in addition to the visible wilting in morphology, *L. barbarum* will also show obvious changes in transcription and metabolism when exposed to heat stress [20].

Compared with other Solanaceae, *Lycium* plants adapt to strong light and arid and saline environments [16–19]. In China, *Lycium* species are also applied as pioneer trees in

vegetation restoration and saline–alkali land improvement, in which they have important ecological value [17]. Their natural growth habitat endows *Lycium* plants with ideal materials for stress biology research; however, the continuing lack of a genome sequence of this genus has severely impeded advances in this field. Recently, a genome sequence for one *Lycium* plant, *Lycium (L.) barbarum*, was released, which undoubtedly will greatly impel studies in *Lycium* biology [18].

To explore the water relationships of *Lycium* plants both under key growth processes and for the abiotic stress response, in the present work, we first identified and characterized AQP family genes from *L. barbarum* at a genome scale and then conducted systematic bioinformatics analysis, including assessments of gene structures, conserved domains, phylogenetic analysis, and cis-elements in promoters, as well as a transcript profiling assay. Our results highlight the importance of these gene families in fresh fruit ripening and seedling heat stress responses and, therefore, lay an invaluable foundation for the in-depth elucidation of the acclimation mechanisms of *Lycium* plants to extreme environments, especially in terms of water transport.

## 2. Materials and Methods

### 2.1. Plant Materials

The different tissues, including roots, stems, leaves, and flowers, as well as the five ripening stages of *Lycium (L.) barbarum* (Ningqi 7, N7), were collected from three of the 5-year-old trees at the Wolfberry (Lycium) Germplasm Repository of Ningxia, Academy of Agriculture and Forestry Sciences, Ningxia Hui Autonomous Region, China (38°080′ N, 106°090′ E and altitude 1100 m). To assay fruit ripening dynamics, fruits were sampled at five different stages (S1, S2, S3, S4, and S5) under their natural state, as described by Cao et al. (2021) [18]. Within 30 days before sampling, the mean high temperature and low temperature of Yinchuan were 27.50 °C and 12.37 °C, respectively, while the field was regularly irrigated to keep soil moisture suitable. In addition to rational field management, healthy fruits in good condition were selected after sampling and used for experiments. Our field studies were conducted in accordance with local legislation and appropriate permissions.

### 2.2. Heat Stress Assay

Uniform clonal seedlings that were grown in a greenhouse for approximately 5 weeks were transported to a growth chamber. After acclimation to the artificial environment (25 °C) for one week, the seedlings were divided into two parts. Whereas one part was still left in the same growth chamber as the control, the other part was transported to a 42 °C growth chamber for heat stress exposure. The seedlings were subjected to heat stress for 0, 1, 3, 6, 12, and 24 h. The leaves were sampled at each time point and divided into three biological replicates, then immediately frozen by liquid nitrogen for RNA extraction.

### 2.3. Relative Water Content Determination of L. barbarum Fresh Fruits

After weighing their fresh weight, the fruits (5~6) at different ripening stages (three biological replicates per stage) were transferred from their aluminum boxes to the corresponding centrifuge tubes for water soaking. The amount of water added to each tube was consistent and the fruits were completely immersed. After 24 h, the fruits were taken out, the surface moisture was removed, and the fruits were transferred into their corresponding aluminum boxes to weigh the saturated weight. Then, the aluminum boxes containing the chopped fruits were transferred to an oven for drying. The samples were first dried at 60~80 °C for 2~3 h, ensuring that the tissue became brittle and dry, and then dried at 100~105 °C for 1~2 h. After the difference between the two weights was less than 0.002 g, the weight was denoted as the dry weight. The RWC was calculated by the formula RWC (%) = (fresh weight − dry weight) × 100/(saturated weight − dry weight).

### 2.4. Identification and Chromosomal Location of LbAQP Genes

To identify the putative AQP proteins in *L. barbarum* (LbAQPs), the 35 AtAQPs in the *A. thaliana* genome were downloaded from the TAIR database (https://www.arabidopsis.org/) and were then used as queries to BLAST search the *L. barbarum* genome (https://www.ncbi.nlm.nih.gov/genome/81199?genome_assembly_id=1656998, acessed on 20 August 2022) with an E-value of $e^{-10}$. Then, preliminary amino acid sequences that may have the function of LbAQPs were obtained according to the homology of AtAQPs. On this basis, conservative AQP domains obtained from the PFAM database (http://pfam-legacy.xfam.org/) were blast searched against these candidate sequences. Finally, amino acid sequences without conserved AQP domains and redundant sequences were manually removed.

The molecular weights (MWs), isoelectric points (pIs) and grand average hydropathicity (GRAVY) values of the LbAQP proteins were analyzed with ProtParam (http://web.expasy.org/protparam/) [21].

We retrieved the genome annotation files (for internal use only) from the *L. barbarum* genome database of NCBI and summarized their physical positions into a graph using TBtools software v1.098774, in which the chromosome numbers and positions of each sequence in the genome were indicated.

### 2.5. Classification of LbAQP Protein Members and Construction of a Phylogenetic Tree

The phylogenetic tree was constructed by MEGAX software from a ClustalX alignment of related amino acid sequences (bootstrap replicates = 1000) using the maximum likelihood method. The ML tree was formatted for visualization by the Chiplot website (https://www.chiplot.online/).

### 2.6. Structure and Conserved Motif Analysis of LbAQPs

Sequence alignments between selected sequences and the genome were carried out according to the GFF format genome annotation files obtained from the genome database in NCBI, and the intron-exon structure information of these genes was generated. TBtools software was used to draw the structure map. Conserved motif analysis was performed in the classic mode of Multiple Em for Motif Elicitation (MEME, https://meme-suite.org/meme/), where the number of motifs was set to 15, the E-value was set to $e^{-10}$, and the other settings were consistent with the default parameters.

### 2.7. Promoter Cis-Element Analysis of LbAQPs

The promoter regions 1 kb upstream of the corresponding genes were analyzed, and the cis-elements were predicted by the PlantCARE database (https://bioinformatics.psb.ugent.be/webtools/plantcare/html/). Subsequently, according to the results of the PlantCARE calculations, we recorded the numbers of cis-elements in these sequences and summarized this information into a figure for subsequent analysis.

### 2.8. Quantitative Real-Time PCR (qRT-PCR)

Total RNA was isolated from the indicated samples using TRIzol reagent (Invitrogen), and its quality was determined by a NanoDrop (Thermo Scientific, Wilmington, DE, USA). First-strand cDNAs were synthesized from 0.5 µg of RNA using the Super-Script III First-Strand Synthesis SuperMix Kit (Invitrogen, Grand Island, NY, USA). qRT-PCR was conducted using 0.01 µg of the cDNA on a LightCycler 480 Instrument System (Roche, Diagnostics GmbH, Mannheim, Germany) with KAPA SYBR FAST qPCR Master Mix and with an initial denaturing step at 95 °C for 5 min, followed by 55 cycles of 95 °C for 10 s, 60 °C for 20 s, and 72 °C for 5 s. The fold changes in the relative expression levels were analyzed via the $2^{-\Delta\Delta CT}$ method using the LbACTIN1 gene as an internal control. The gene-specific primers used for qRT-PCR are listed in Table S1. All experimental results were performed with three biological and technical replicates.

*2.9. Statistical Analysis*

Statistical analyses were performed using SPSS version 19.0. Parameter differences among various ripening stages of fruits were determined using one-way ANOVA with appropriate post hoc analysis.

**3. Results**

*3.1. Characterization of the LbAQP Gene Family*

To identify *LbAQP* family genes in the *L. barbarum* genome, the 35 Arabidopsis AQP protein sequences and the MIP PF00230 conserved domain were employed as queries to search against the *L. barbarum* genome database using the BlastP program. Forty-seven genes were identified by homologous alignment, and nine genes were eliminated by conserved domain and amino acid site analysis. Finally, 38 full-length genes encoding potential *LbAQPs* were identified and named according to their sequence similarity and phylogenies with both individual AtAQP proteins and SlAQP proteins (Table 1). While three of these genes (*LbPIP1;5*, *LbPIP2;7*, and *LbNIP4;3*) encoded severely truncated proteins, three genes experienced clear gene duplication events (*LbTIP2;2*, *LbNIP1;2*, and *LbNIP6;1*). In-depth analysis of the identified 38 *LbAQPs* was performed to determine their CDS, TMHMM, isoelectric point (pI), MW, and grand average of hydropathicity (GRAVY). Most of the LbAQPs consisted of CDSs ranging in length from 740 to 900 base pairs (bp).

**Table 1.** Detailed information on 38 aquaporin (AQP) genes of *Lycium* (*L.*) *barbarum* and their encoded proteins.

| Gene Name | Gene ID | CDS Length (bp) | Size (aa) | Size (aa) | PI | TMHMM | GRAVY |
|---|---|---|---|---|---|---|---|
| *LbPIP1;1* | Lba07g01359 | 861 | 287 | 30.777 | 8.31 | 6 | 0.409 |
| *LbPIP1;2* | Lba06g02040 | 855 | 285 | 30.784 | 8.64 | 6 | 0.318 |
| *LbPIP1;3* | Lba04g00268 | 858 | 286 | 30.694 | 7.69 | 6 | 0.375 |
| *LbPIP1;4* | Lba09g02427 | 861 | 287 | 30.866 | 7.68 | 6 | 0.422 |
| *LbPIP1;5* | Lba09g02372 | 492 | 164 | 17.854 | 9.91 | 3 | 0.746 |
| *LbPIP1;6* | Lba01g01166 | 858 | 286 | 30.627 | 9.10 | 6 | 0.416 |
| *LbPIP2;1* | Lba03g00307 | 849 | 283 | 30.223 | 8.21 | 6 | 0.501 |
| *LbPIP2;2* | Lba03g00306 | 849 | 283 | 30.168 | 6.94 | 6 | 0.496 |
| *LbPIP2;4* | Lba08g00456 | 861 | 287 | 30.732 | 6.37 | 6 | 0.537 |
| *LbPIP2;5* | Lba08g01659 | 855 | 285 | 30.425 | 8.24 | 6 | 0.408 |
| *LbPIP2;6* | Lba06g00111 | 861 | 287 | 30.624 | 8.57 | 6 | 0.590 |
| *LbPIP2;7* | Lba01g02704 | 528 | 176 | 18.901 | 5.16 | 4 | 0.545 |
| *LbPIP2;8* | Lba06g03476 | 852 | 284 | 30.435 | 9.28 | 6 | 0.486 |
| *LbPIP2;9* | Lba08g01171 | 849 | 283 | 30.150 | 9.24 | 6 | 0.512 |
| *LbPIP2;10* | Lba10g01287 | 852 | 284 | 30.421 | 8.83 | 6 | 0.545 |
| *LbPIP2;11* | Lba12g01911 | 810 | 270 | 28.867 | 8.82 | 6 | 0.522 |
| *LbPIP2;12* | Lba05g00065 | 1029 | 343 | 37.897 | 6.40 | 5 | 0.285 |
| *LbTIP1;1* | Lba01g02671 | 753 | 251 | 25.712 | 5.16 | 6 | 0.740 |
| *LbTIP2;1* | Lba07g01176 | 744 | 248 | 25.043 | 6.15 | 7 | 0.988 |
| *LbTIP2;2* | Lba03g02891 | 1476 | 492 | 49.630 | 5.51 | 15 | 1.009 |
| *LbTIP2;3* | Lba01g01487 | 750 | 250 | 25.220 | 5.35 | 6 | 0.919 |
| *LbTIP2;4* | Lba01g02017 | 744 | 248 | 24.988 | 5.66 | 7 | 0.952 |
| *LbTIP3;1* | Lba01g02391 | 774 | 258 | 27.175 | 6.70 | 6 | 0.621 |
| *LbTIP3;2* | Lba03g01980 | 780 | 260 | 27.752 | 8.07 | 6 | 0.523 |
| *LbTIP4;1* | Lba04g01403 | 741 | 247 | 25.883 | 6.01 | 7 | 0.866 |
| *LbTIP5;1* | Lba03g01262 | 768 | 256 | 26.549 | 8.58 | 6 | 0.709 |
| *LbNIP1;2* | Lba12g01378 | 1614 | 538 | 57.114 | 8.43 | 11 | 0.431 |
| *LbNIP2;1* | Lba11g02541 | 753 | 251 | 26.644 | 9.20 | 5 | 0.381 |
| *LbNIP3;1* | Lba01g02539 | 1041 | 347 | 37.614 | 8.45 | 6 | 0.386 |
| *LbNIP3;2* | Lba07g01497 | 810 | 270 | 29.075 | 9.30 | 5 | 0.564 |
| *LbNIP4;1* | Lba12g02217 | 834 | 278 | 29.332 | 5.92 | 7 | 0.698 |
| *LbNIP4;2* | Lba05g02002 | 2859 | 953 | 105.105 | 7.34 | 5 | -0.546 |
| *LbNIP4;3* | Lba12g00430 | 441 | 147 | 15.815 | 9.61 | 3 | 0.678 |
| *LbNIP4;4* | Lba12g00428 | 837 | 279 | 29.702 | 7.55 | 6 | 0.702 |
| *LbNIP5;1* | Lba09g02501 | 1167 | 389 | 40.914 | 6.11 | 5 | 0.249 |
| *LbNIP6;1* | Lba03g02537 | 1866 | 622 | 64.687 | 8.48 | 12 | 0.421 |
| *LbXIP1;2* | Lba08g01139 | 975 | 325 | 34.597 | 7.66 | 7 | 0.698 |
| *LbXIP1;6* | Lba06g03416 | 972 | 324 | 34.684 | 7.04 | 6 | 0.737 |

In addition, nine sequences were longer than 900 bp, among which seven sequences were longer than 1000 bp and one sequence was longer than 2000 bp. The predicted proteins ranged in length from 147 to 953 amino acids, with 15.815 to 105.105 kDa MW. The pI values varied between 5.16 and 9.91. The GRAVY values were also detected through bioinformatics analysis and were positive, except for LbNIP4;2, and ranged from −0.546 to 1.009. The conserved transmembrane domains (TMDs) values ranged from 3 to 15. The TMDs were predicted using TMHMM version 2.0 (http://www.cbs.dtu.dk/services/TMHMM/).

### 3.2. Physical Distribution of LbAQPs on L. barbarum Chromosomes

We further analyzed the *LbAQP* gene location on the *L. barbarum* chromosomes (Chr). It was shown that the *LbAQP*-encoded genes were unevenly distributed on 11 out of 12 *L. barbarum* chromosomes, with the exception of chr 02. In detail, there were seven, six, two, two, four, three, four, three, one, one, and five genes located on chromosomes 01, 03, 04, 05, 06, 07, 08, 09, 10, 11, and 12, respectively (Figure 1).

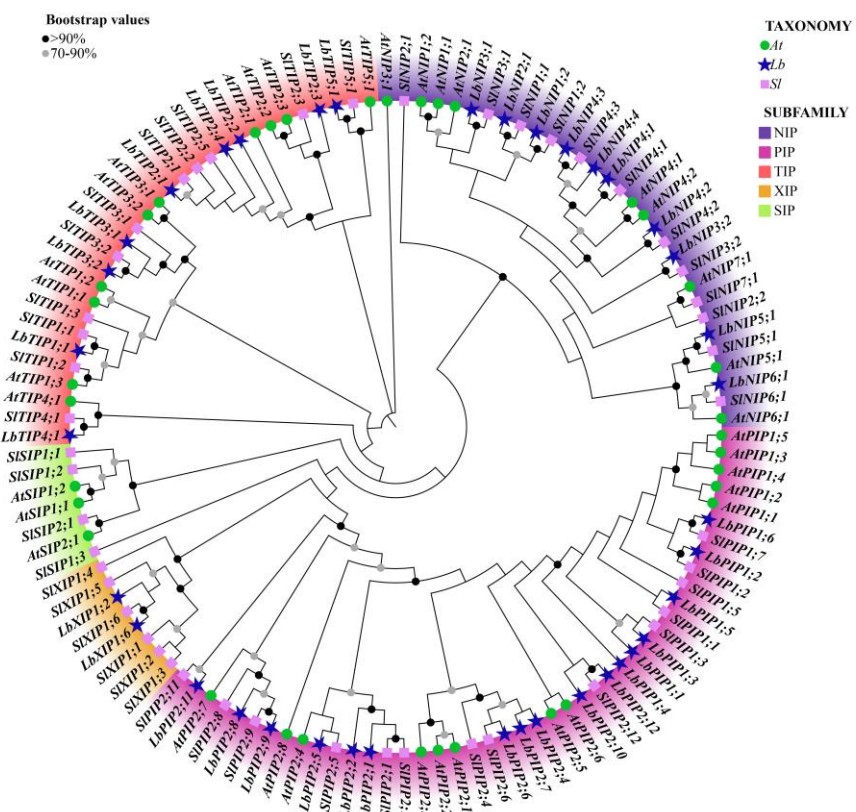

**Figure 1.** Phylogenetic analysis of the 38 *Lycium barbarum* AQPs (LbAQPs) with *Arabidopsis* and tomato homologs. Deduced amino acid sequences were aligned using ClustalX, and a phylogenetic tree was constructed using the bootstrap maximum likelihood tree (1000 replicates) method and MEGAX software. The full-length AQP protein sequences, including 38 members from *Lycium barbarum* (Lb), 35 members from *Arabidopsis thaliana* (At), and 47 members from *Solanum lycopersicum* (Sl), were classified into PIP, TIP, NIP, XIP, and SIP subfamilies, respectively. The branches of different classes have altered colors, and each represents a different subfamily.

### 3.3. Phylogenetic Comparison of LbAQP Proteins of L. barbarum, Tomato, and Arabidopsis

To study the evolutionary characteristics of genes and the evolutionary relationships among AQP proteins, we performed cross-genus phylogenetic analysis with 38 LbAQPs, along with 35 and 45 AQP proteins from *Arabidopsis* and tomato, respectively (Figure 2). According to the known *Arabidopsis* and tomato AQP families, LbAQPs can be divided into four distinct subfamilies: LbPIPs, LbTIPs, LbNIPs, and LbXIPs. There are 17 *LbPIP*, 9 *LbTIP*, 10 *LbNIP*, and 2 *LbXIP* in the *L. barbarum* genome, respectively. In contrast to

*Arabidopsis* and other *Solanaceae* species [21–23], there are no LbSIPs found in the *L. barbarum* genome. The *17 LbPIPs* can be further subdivided into two subgroups, 6 *LbPIP1* and 11 *LbPIP2*. The comparative phylogenetic trees of AQP subfamilies among *Arabidopsis*, tomato, and *L. barbarum* demonstrated that AQP was highly species-specific. We also found that the proteins in the same or adjacent classification groups belonged to the same subfamily, which further confirmed the phylogenetic tree analysis.

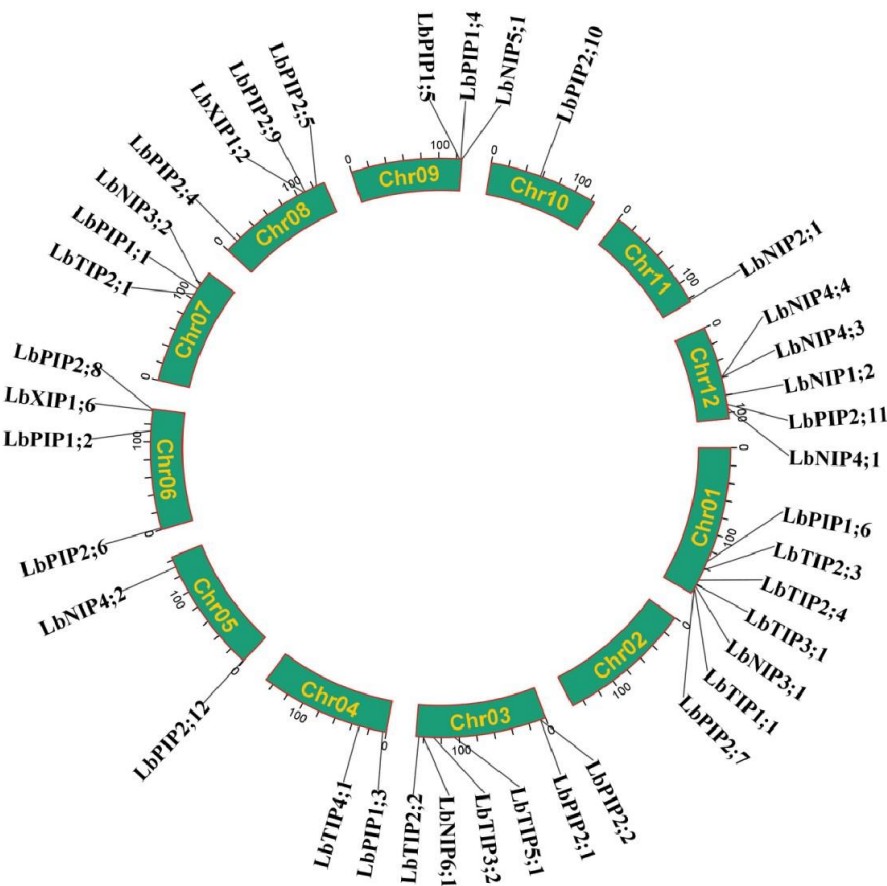

**Figure 2.** Chromosomal distribution of *LbAQP* genes.

*3.4. Gene Structure, Conserved Motifs, and Phylogenetic-Tree-Based Classification of LbAQPs*

Multisequence alignment was performed to explore the evolutionary relationships of all 38 *LbAQP* genes (Figure 3A). A maximum likelihood tree was constructed by comparing related amino acid sequences. We also used MEME software to analyze the conserved motifs of LbAQPs with motif-specific sequences. Full-length protein sequence analysis identified a total of 10 motifs (Figure 3B). The number of conserved motifs in each LbAQP varied between three and eight. Although the conserved motifs of the 38 LbAQP genes were different in composition, they all contained motif 2, motif 3, and motif 4 and they were arranged in the same order, with motif 2 first, then motifs 3 and 4. Motifs 2, 3, and 4 existed in most LbAQPs, representing the characteristic structures of AQP. Motif analyses showed that most of the motifs were specific to subfamilies. Most members of the LbPIPs contained motifs 2, 3, 4, 5, 6, 8, 9, and 10, with the exception of LbPIP1;5, LbPIP2;7, and PIP2;12. All members of the LbTIPs, except LbTIP5;1, contained motifs 1, 2, 3, 4, and 7. While motifs 1, 2, 3, 4, and 5 appeared in two LbXIPs, LbNIPs contained motifs 1, 2, 3, and 4, except for LbNIP4;3. There was a clear duplication of motif 1 in subfamilies LbXIPs and LbNIPs, with two copies present in their genes in these two subfamilies.

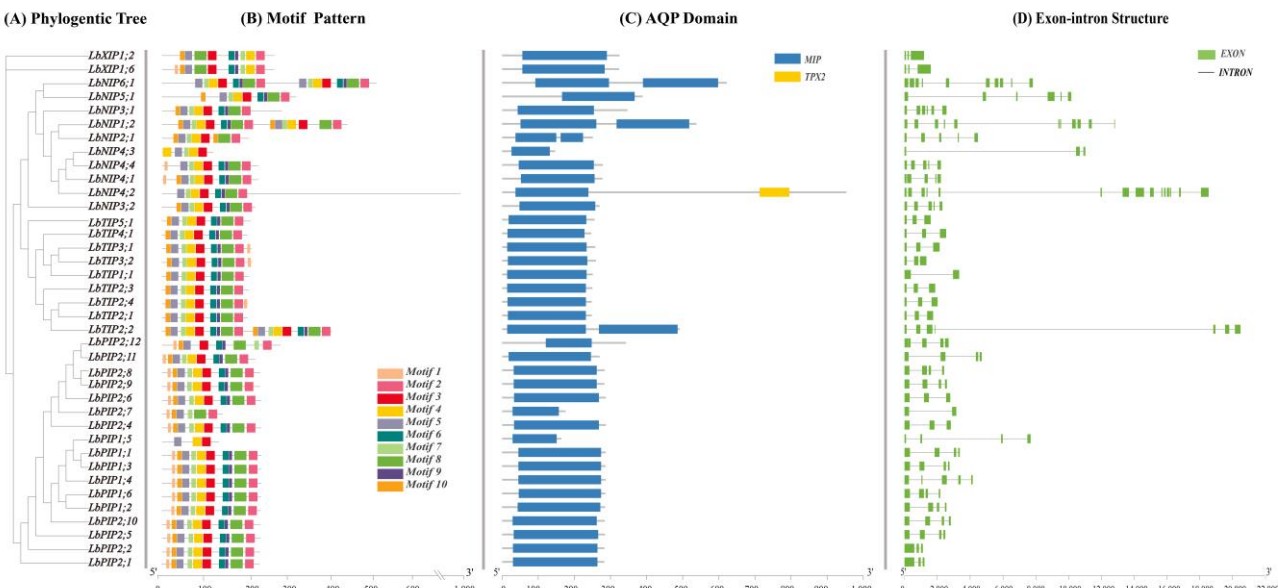

**Figure 3.** Phylogenic relationships, gene structure conserved domains, and conserved motif analyses. (**A**) A phylogenetic tree of *LbAQP* proteins was constructed with MEGAX software. (**B**) Conserved motif distribution of *LbAQP* proteins. The conserved motifs identified with MEME are displayed with boxes in different colors. A total of 10 motifs were identified. The scale at the bottom shows the length of the protein. (**C**) Predicted conserved structural domains of *LbAQP* proteins. Gray lines represent the length of each protein sequence, and conserved domains are indicated by colored boxes. (**D**) Exon-intron structures of the *LbAQP* genes. Gene structures were analyzed with Tbtools software. Exons and introns are indicated with green boxes and gray lines, respectively. The lengths of the exons and introns (bp) are indicated on the *x*-axis. The combined figure was illustrated with Tbtools software.

While 34 members out of 38 LbAQPs have one conserved MIP domain (Figure 3C), *LbTIP2;2*, *LbNIP1;2*, and *LbNIP6;1* have two similar MIP domains due to gene duplication. In addition to containing one MIP domain, *LbNIP4;2* also contained a targeting protein for Xklp2 (TPX2) domain.

To better understand the corresponding gene structure, the exon-intron structures of 38 *LbAQPs* were analyzed by the Pfam database based on amino acid sequences (Figure 3D). Our study revealed that the number and length of introns were significantly different among different subfamilies in LbAQPs, in which the number ranged from one to five introns. Eleven *LbPIP* genes had three introns, while four genes (*LbPIP2;1*, *LbPIP2;2*, *LbPIP2;4*, and *LbPIP2;6*) had two introns, one gene (LbPIP1;4) had four introns, and one gene (*LbPIP2;7*) had one intron. Seven LbTIP genes had two introns, *LbTIP1;1* had one, and *LbTIP2;2* had three. Six LbNIP genes had four introns, *LbNIP3;1* and *LbNIP5;1* had five introns, and *LbNIP4;3* had two introns. Both *LbXIP* genes had two introns. The lowest number of introns was observed in *LbTIPs* (~two) and *LbXIPs* (two), followed by *LbPIPs* (~three) and NIPs (~4). The varied number of introns among the *LbAQPs* contributed to the variations in gene length (Figure 3).

Multiple sequence alignment of the DNA-binding domains of 38 LbAQP proteins revealed the conserved amino acid sequences of AQP (Figure 4). Most of the LbAQPs contained dual NPA motifs, except *LbPIP2;7*, *LbPIP2;12*, *LbPIP1;5*, and *LbNIP4;3*, which were found to harbor a single NPA motif. Both the first and second NPA motifs were found to be conserved in the LbPIP and LbTIP subfamilies. While LbNIP2;1 and LbNIP5;1 showed alanine to serine substitutions, *LbNIP4;3* showed alanine to valine substitutions in the first NPA motif. It was found that valine substituted alanine in the second NPA motif for both NIP5;1 and NIP6;1. Two LbXIPs were not conserved in the first NPA motif, in which valine substituted alanine for *LbXIP1;2*, while serine substituted asparagine and

valine substituted alanine for *LbXIP1;6*. Therefore, both the first and the second NPA motifs were not conserved for the LbNIP subfamily.

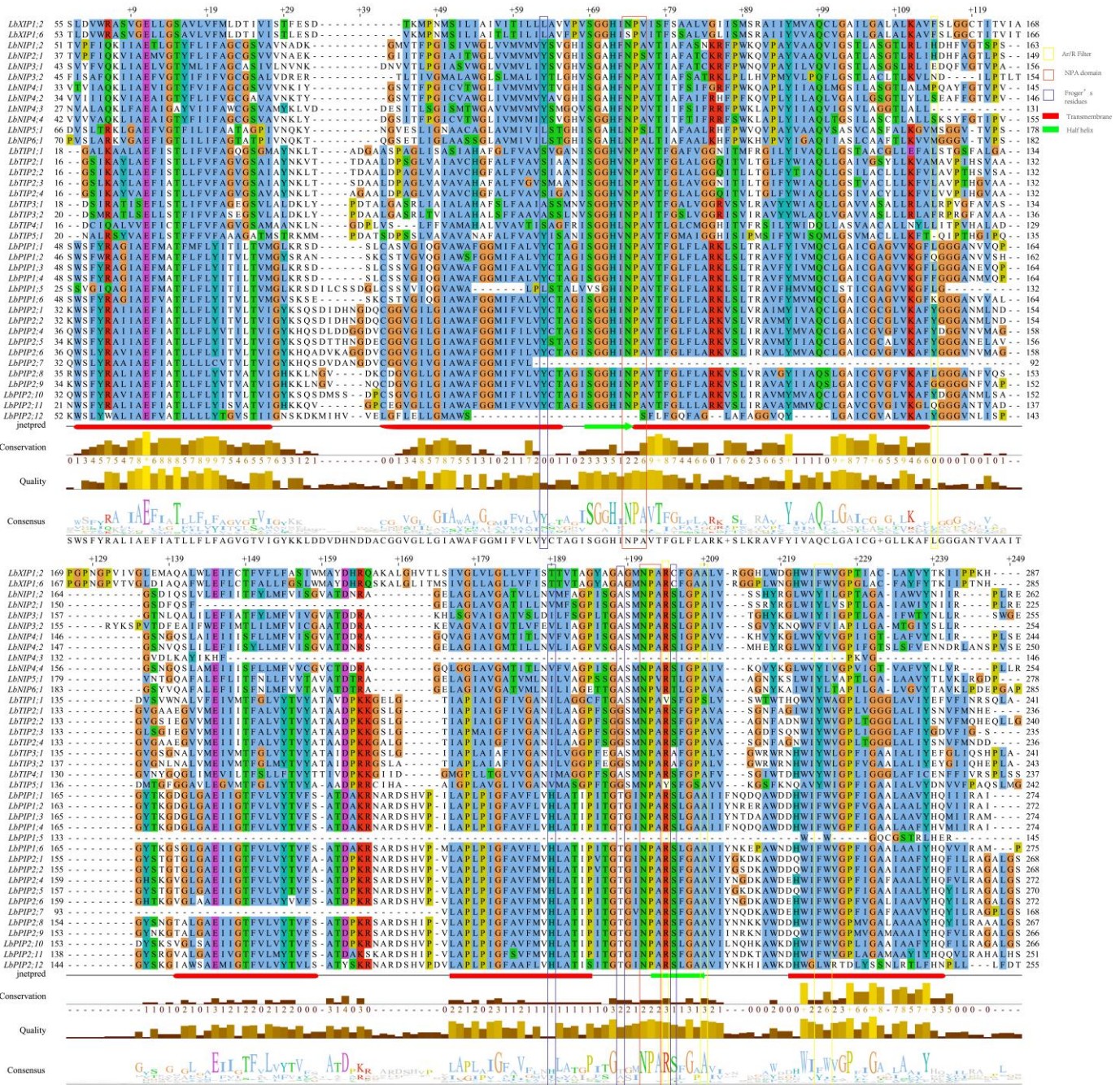

**Figure 4.** Protein sequence alignment of LbAQPs. Conserved transmembrane domains (TM1-6) and amino acids at NPA domains, ar/R selectivity filters, and Froger's residues identified in LbAQPs.

### 3.5. Analyzing Cis-Elements in the LbAQP Promoters

Promoter cis-acting elements are important binding regions of transcription initiation factors and play an important role in regulating gene expression. To further explore the possible biological functions of the 38 LbAQPs, the presence of cis-acting elements in the 1 kb upstream promoter regions of the corresponding genes was predicted using the Plant-CARE database (Figure 5). This result indicated that, except for the core promoter elements, such as TATA-box and CAAT-box (data not shown), some unique cis-acting elements related to hormones, stress resistance, and tissue and organ development were identified,

which are speculated to play regulatory roles in the activation and induction of *LbAQP* expression. The identified phytohormone responsiveness elements included ABA response-related element (ABRE), TGACG-motif (cis-acting regulatory element involved in MeJA responsiveness), Eth-responsive element (ERE), P-box and GA-responsive element (GARE) motif, TCA-element (SA-acting element), and TGA-element (auxin-responsive element). The biotic/abiotic stress response elements included MYC, MYB, stress response element (STRE), LTR (low-temperature control element), wound-responsive element (WRE3), W-box, TC-rich repeats (cis-acting element involved in defense and stress responsiveness), DRE core (drought and osmotic stress induction element), anaerobic induction regulatory element (ARE), and MYB-binding site (MBS, drought-responsive element), which may be related to the tolerance and response mechanisms of plants to biotic or abiotic stress. The others included G-box (light-responsive element), Box-4 (part of a conserved DNA module involved in light response), TCT-motif (part of a light-responsive element), AAGAA-motif (cis-elements for oxidative defense pathway), and circadian (cis-acting regulatory element involved in circadian control). Among these, the more abundant cis-elements in the LbAQP promoters were MYC, MYB, ABRE, G-box, ARE, STRE, and Box-4, reflecting the vital functions of these genes in abiotic stress response and light response.

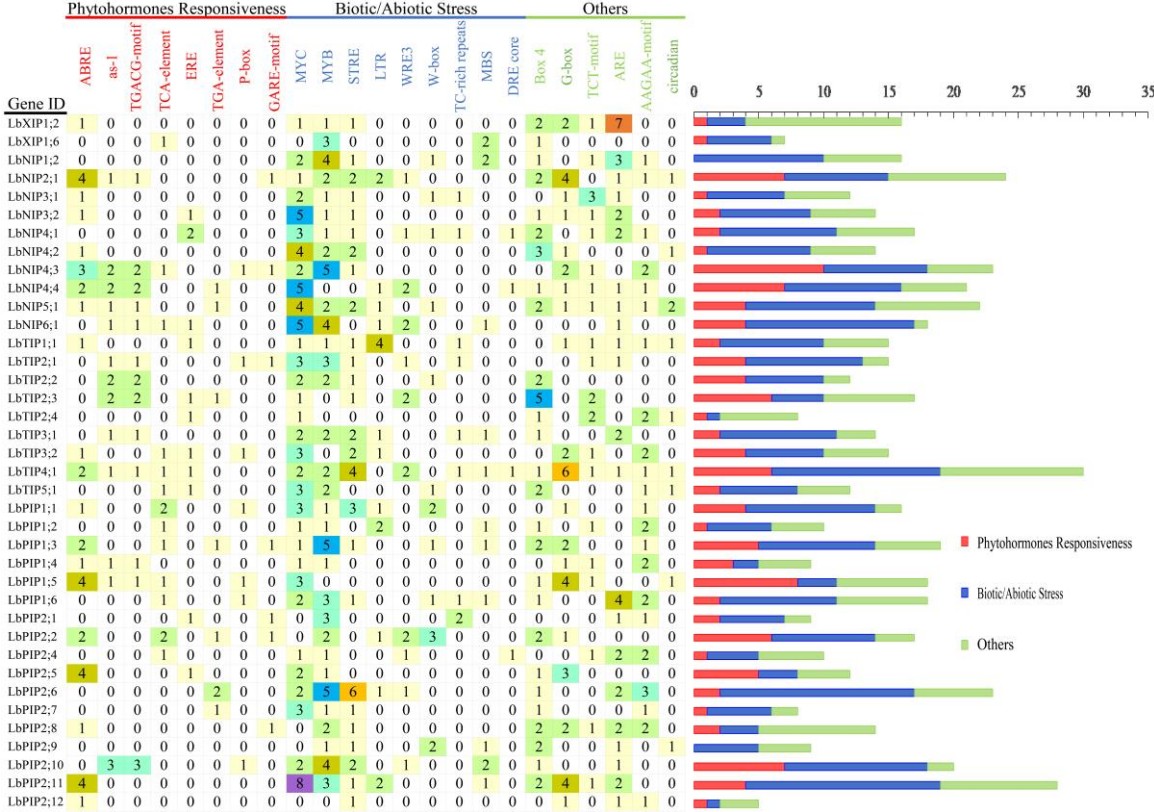

**Figure 5.** The cis-acting elements on the putative promoters of the *LbAQP* genes. The identified cis-acting elements were mainly divided into three major categories: phytohormone responsiveness, biotic/abiotic stress, and others.

### 3.6. Tissue-Specific Expression

To determine the tissue-specific expression of this family of genes, a quantitative real-time PCR (qRT-PCR) assay was conducted in various tissues of *L. barbarum*, including roots, stems, leaves, flowers, and fruits (Figure 6). Five genes were specifically expressed in roots (*LbNIP3;1*, *LbTIP2;3*, *LbTIP2;4*, *LbTIP2;2*, and *LbPIP1;4*). Although they were also expressed in other tissues, the genes with the highest transcripts in roots are *LbNIP4;2*, *LbPIP1;3*, *LbPIP1;1*, *LbPIP2;6*, *LbPIP2;4*, *LbNIP1;2*, and *LbNIP2;1*. The genes *LbTIP3;1* and

*LbTIP3;2* were mainly expressed in fruits. Eleven genes had maximal transcript levels in stems versus other tissues, which included *LbNIP5;1*, *LbNIP6;1*, *LbTIP4;1*, *LbTIP2;1*, *LbXIP1;6*, *LbPIP1;6*, *LbPIP1;2*, *LbPIP2;8*, *LbPIP2;11*, *LbPIP2;7*, and *LbPIP2;10*. The genes that were highly coexpressed in stems and leaves were *LbTIP2;1*, *LbXIP1;2*, *LbPIP2;7*, *LbPIP2;10*, *LbPIP2;1*, *LbNIP6;1*, and *LbPIP2;2*, while the genes that were highly coexpressed in roots and stems were *LbNIP5;1*, *LbNIP2;1*, *LbNIP1;2*, and *LbPIP2;6*. The genes that were coexpressed in roots, stems, and leaves were *LbNIP1;2* and *LbPIP2;7*. Meanwhile, the gene that was specifically expressed in leaves was *LbPIP2;5* and the genes with the highest transcript levels in leaves were *LbXIP1;2* and *LbPIP2;12*. The most highly expressed genes in flowers were *LbNIP4;3*, *LbNIP4;1*, and *LbTIP5;1*, although many genes had their lowest transcript levels in flowers vs. other tissues, such as *LbPIP1;3*, *LbPIP1;1*, *LbPIP2;9*, and *LbPIP2;6*. Taken together, the overlapping and preferential expression patterns of these *LbAQPs* might confer *Lycium* to conduct distinct ABA responses in specific tissues.

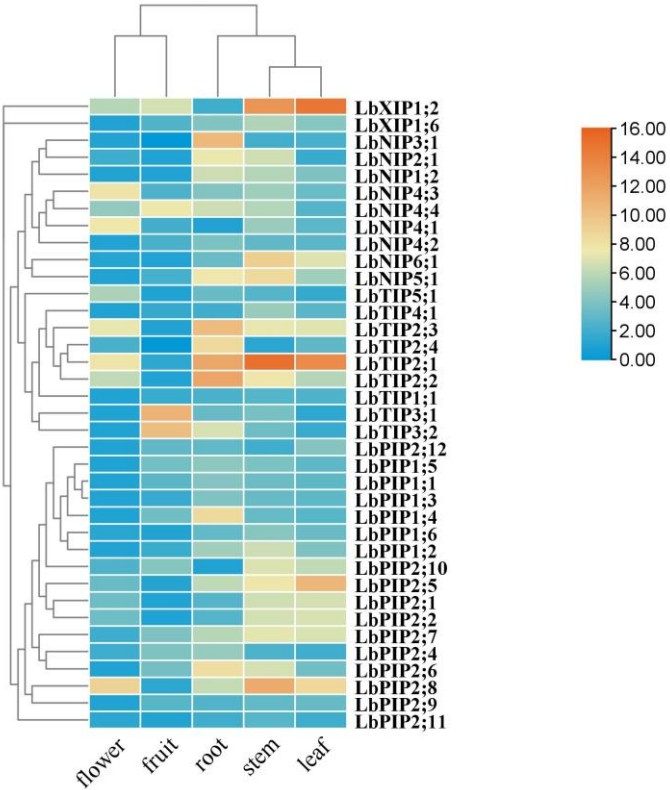

**Figure 6.** Tissue-specific expression of *LbAQP* genes. Tissue-specific expression of *LbAQPs* was determined by quantitative real-time PCR (qRT-PCR) in leaves, young roots, stems, ripening fruits, and flowers with gene-specific primers. qRT-PCR was performed in triplicate, and the fold changes were analyzed via the $2^{-\Delta\Delta CT}$ method using the *LbACTIN1* gene as an internal control. Values are the means of three independent experiments.

### 3.7. Expression Profiles of LbAQPs during Fruit Ripening

Fruit development and ripening are complex processes that undergo dramatic physiological changes, including a changing water status [24–26]. To determine whether the expression of *LbAQPs* is sensitive to developmental cues, the transcriptional levels of 38 *LbAQPs* were analyzed using qRT-PCR with gene-specific primers (Figure 7). Within the *LbAQP* gene families, 24 genes were downregulated and three genes (*LbNIP4;1*, *LbNIP4;3*, and *LbNIP4;4*) were upregulated during *L. barbarum* fruit ripening. While the upregulated genes all belonged to the LbNIP subfamily, the downregulated genes were distributed in all four subfamilies. Ten *LbAQP* genes presented irregular expression patterns, in which the transcripts of four genes first increased and then decreased, whereas those of six genes first decreased and then increased during fruit ripening. Among the downregulated genes,

most were significantly expressed in the S1 stage, presenting a negative correlation with the fruit relative water content (RWC).

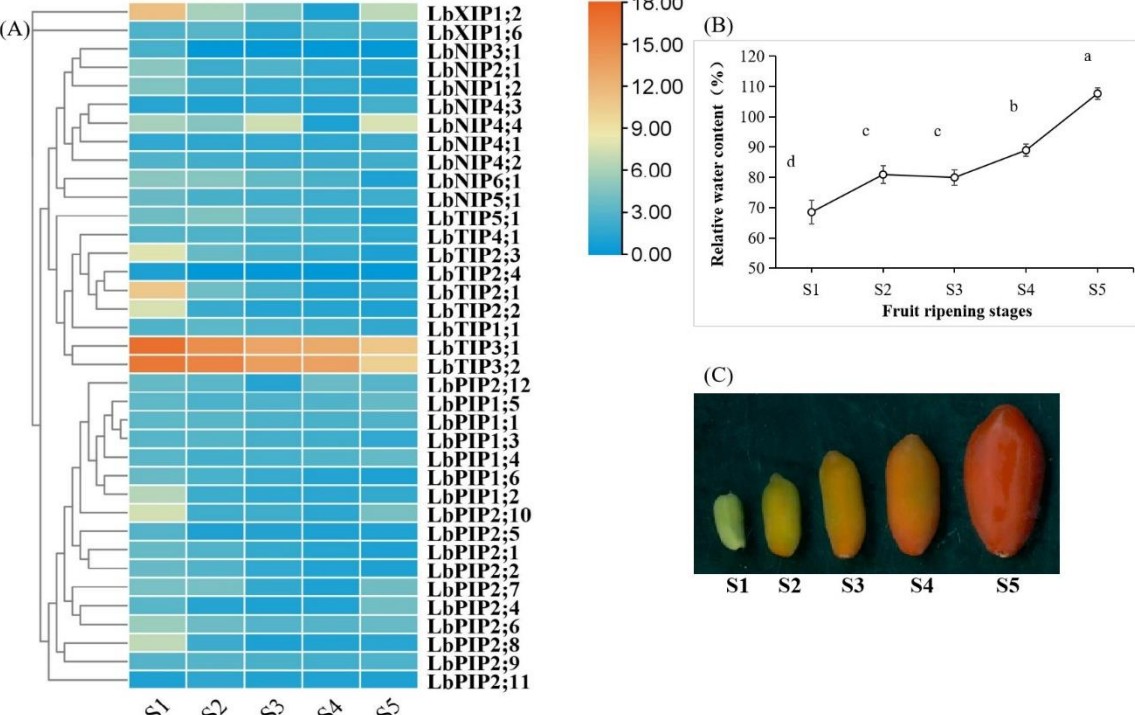

**Figure 7.** Development-dependent expression profiling during fruit ripening. (**A**) The transcript abundance of *LbAQPs* during fruit ripening was analyzed at five different developmental stages under their natural state. The fold changes in the relative expression levels were analyzed via the $2^{-\Delta\Delta CT}$ method using the *LbACTIN1* gene as an internal control. Values are the means of three independent experiments. (**B**) Relative water content (%) of the fruits for the above five ripening stages. Values represent the mean $\pm$ SEM, n = 3 (biological replicates). Means with different letters are significantly different ($p < 0.01$; one-sided ANOVA). (**C**) Phenotypes of the five representative ripening stages (S1, S2, S3, S4, and S5) of *L. barbarum* fruits.

### 3.8. Expression Profiles of LbAQPs in Response to Heat Stress

Whereas most of the *LbAQP* genes were significantly upregulated by heat stress, *Lb-NIP5;1*, *LbXIP1;2*, and *LbPIP1;6* were downregulated (Figure 8). For the upregulated genes, most of their transcript abundance presented first increasing and then decreasing expression patterns, while several were maintained at constant levels (*LbNIP4;2* and *LbTIP1;1*), several continuously increased (*LbNIP3;1*, *LbTIP5;1*, and *LbXIP1;6*), and several had irregular changes during 24 h of heat stress. Compared with the control values, approximately half of the gene transcripts were upregulated by over fivefold. Among them, *LbNIP3;1*, *LbTIP3;1*, *LbTIP2;4*, and *LbNIP4;4* were upregulated by over 15-, 30-, 40-, and 70-fold, respectively. Six genes (*LbNIP4;1*, *LbNIP4;4*, *LbTIP3;1*, *LbTIP3;2*, *LbPIP2;9*, and *LbPIP2;11*) achieved maximal transcription at 1 h, and 13 genes peaked at 3 h, which can be considered earlier response genes. The others were part of the later response genes.

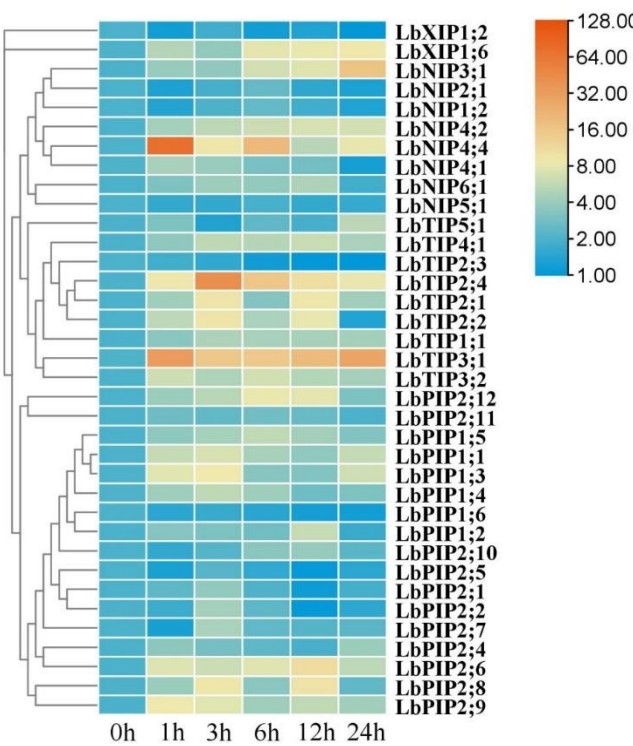

**Figure 8.** Expression profiling responses to heat stress. The expression levels of *LbAQPs* were measured in the leaves of 2-month-old *L. barbarum* subjected to 42 °C for 0, 1, 3, 12, or 24 h. The fold changes in the relative expression levels were analyzed via the $2^{-\Delta\Delta CT}$ method using the *LbACTIN1* gene as an internal control. Values are the means of three independent experiments.

## 4. Discussion

Thirty-eight homologs of *Arabidopsis* AQPs were identified in the *L. barbarum* genome, indicating this gene family conservation in higher plants [6]. Certainly, it should be noted that clear genus-specific features exist. Within *Solanaceae* species, 45, 47, and 76 AQPs have been identified in tomato (*Solanum lycopersicum*), potato (*Solanum tuberosum*), and tobacco (*Nicotiana tabacum*) genomes, respectively [21–23]. Moreover, all five subfamilies of AQPs that have been identified thus far were present in the above three Solanaceae species, whereas SIP was not found in the *L. barbarum* genome. Interestingly, within the 38 *LbAQP* genes, both truncation and duplication events occurred. It is increasingly recognized that not all plant AQP subfamilies are good water channels. Within the NIPs (mainly), as well as some PIPs and XIPs, transported substrates include metalloids, protonated organic acids, or metal complexes [27]. The polymorphism of LbAQPs may reflect the multifunctionality of this family of proteins and may also reflect the best acclimation of *Lycium* plants to extreme environments, reversing other Solanaceae species. Of course, it cannot be ruled out that now we only have a relatively coarse reference genome of *L. barbarum*. In the future, with the further improvement of the genome map, the number of AQP genes for *Lycium* plants at the whole-genome level may also vary, as has been shown in maize. Moreover, it should be highlighted that different species within *Lycium* plants may have varied AQP gene numbers, similar to what has been observed in cotton, olive trees, and *Linum* species. Pangenome exploration will bring interesting results in the future.

Developing fruits are strong sink organs, and the accumulation of sugars in them causes a negative water potential, which endows AQPs with pivotal roles at both the tissue and cellular levels [3,28,29]. An attention study was conducted in tomato fruits, in which regulation of AQP expression can clearly modify fruit quality (e.g., size, flavor, nutrition, and firmness) by the method of deficit irrigation-derived water scarcity [3,28,29]. It has been shown in grapes that the discharge of surplus phloem water may be required for normal grape ripening [25]. Our present work showed that most of the AQP genes identified in

*L. barbarum* were downregulated during fruit ripening and presented a negative correlation with fruit RWC. This means that most of them were significantly involved in fruit growth, while several took part in fruit ripening. Considering the coexpression manner of most of the genes and the great change levels within different ripening stages, it can be speculated that at least some AQPs identified here as expressed in fruits are necessary for water transport during fruit development.

With global climate change, extreme temperatures are becoming increasingly frequent, posing serious threats to plant growth and food production [24]. Since they have evolutionarily acclimated to cold and cool local environments for a long time [19], the main threat to current and future global production and quality of *Lycium* plants is heat stress. In the present work, we found that most of the LbAQP transcripts presented quick and sharp responses to heat stress following seedling exposure to a 42 °C temperature, indicating the potential targets of this protein family for engineering the heat tolerance of *Lycium* species. Considering the rapidity and recoverability of the transcriptional response pattern, it is speculated that LbAQPs may also be involved in the heat stress signaling pathway, as has been determined in other physiological processes [1,14]. Among them, *LbNIP4;4* and *LbTIP3;1* may play primary functions, followed by *LbTIP2;4*, *LbTIP2;1*, *LbTIP2;2*, *LbPIP1;3*, *LbPIP2;8*, and *LbPIP1;1*.

Except for transport water, it is increasingly recognized that AQPs have emerged as central membrane targets of environmental and hormonal signaling pathways acting on plant–water relations [1,27,30,31]. Considering that functional compounds, as pharmacodynamic components, are generally secondary metabolites formed by medicinal plants against stress, more studies, including molecular genetics and systems biology approaches, are now needed to comprehend how LbAQPs and secondary metabolites interact during fruit ripening and *Lycium* plants respond to abiotic stress.

## 5. Conclusions

Thirty-eight LbAQP genes were first identified and characterized from the *L. barbarum* genome, which fell into four subfamilies, including 17 LbPIP, 9 LbTIP, 10 LbNIP, and 2 LbXIP. There were no SIP subfamily genes found in *Lycium* plants, unlike in other Solanaceae. The transcript profiling showed that their expression presented clear tissue-, developmental-, and stressor-specific patterns. The rapidity and recoverability of the transcriptional response highlight the potential roles of this protein family in regulating *L. barbarum* fruit ripening and the heat stress response. These findings also suggest that LbPYLs might be good candidates for future biotechnological use to enhance *Lycium* resistance to drought and hot environments. Our results lay a foundation for further studying the molecular mechanism of the water relationship of *Lycium* plants, especially for the two above key physiological processes.

**Supplementary Materials:** The following supporting information can be downloaded at: https://www.mdpi.com/article/10.3390/cimb44120404/s1.

**Author Contributions:** Z.M. and K.Q. designed the research. W.H. and Y.C. performed the experiments. W.H. and Y.Y. conducted the data analyses. K.Q. and X.Q. conducted the field management work. Z.M., M.L. and A.L. wrote the manuscript. All the authors have read and approved the manuscript. All authors have read and agreed to the published version of the manuscript.

**Funding:** This work was jointly supported by the Key Research and Development Project foundation of Ningxia province of China (No. 2020BFH03005), the Foreign Science and Technology Cooperation Project of Ningxia Academy of Agriculture and Forestry Sciences (No. DW-X-2020009).

**Institutional Review Board Statement:** Not applicable.

**Informed Consent Statement:** Not applicable.

**Data Availability Statement:** The original data presented in the study are included in the article/Supplementary Material, further inquiries can be directed to the corresponding author.

**Conflicts of Interest:** The authors declare that the research was conducted in the absence of any commercial or financial relationship that could be construed as a potential conflict of interest.

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
