# Peer review of "Genome-Wide Identification and Expression Analysis of the Aquaporin Gene Family in Lycium barbarum during Fruit Ripening and Seedling Response to Heat Stress"

_cimb, doi:10.3390/cimb44120404_

Round 1

Reviewer 1 Report

The research is novel and well written. However, some minor modifications are needed as per the file attached.

Author Response

Thank you for your comments. We have provided a point-by-point response to your comments as follows (Cover letter).  In addtion,  we upload the revised manuscript (Manuscript-Response) for your review. Please see the attachment.

Cover letter-Response to Reviewer 1 Comments

Point 1:

Rewrite Line:

As elaborate terrestrial living organisms, vascular plants have to maintain their water status as tightly as possible and throughout their life cycle[1].

Response 1: This sentence had been replaced with “ Plant water transport and related molecular components are responsive to an extremely wide array of environmental and hormonal signals, which is an essential process throughout their life cycle[1] ”.

Point 2:

Rewrite Line:

This indicates that the entire XIP subfamily is absent in both Arabidopsis and rice, presenting high variation among plant species.

This line is not clear.

Response 2: This sentence had been replaced with “ In addition, the entire XIP subfamily is absent in both Arabidopsis and rice, presenting its high variation among plant species ”.

Point 3:

Use standard abbreviations.

Response 3: According to the Guidelines for Authors in the MDPI official website, “Define abbreviations the first time they are mentioned in the abstract, text; also the first time they are mentioned in a table or figure” , we had modified our manuscript and marked these revisions up using the “Track  Changes” function.

Point 4: Replace ‘medical plants’ by medicinal plants.

Response 4: We had modified our manuscript and marked the revisions up using the “Track  Changes” function.

Point 5: 

Rewrite Line:  

As one of the world’s most economically and medicinally valuable fruit crops, Goji (枸杞) Lycium of the Solanaceae family contains ~80 species, and these species represent important food plants[16-19].

Response 5: This sentence had been replaced with “ Goji (枸杞) Lycium of the Solanaceae family contains ~80 species, representing important food plants and enjoying the reputation of one of the world’s most economically and medicinally valuable fruit crops ”.

Reviewer 2 Report

The work detailed in this manuscript is a good attempt but it seems like “Expression Analysis of the Aquaporin Gene Family in Lycium barbarum During Fruit Ripening” and “Seedling Response to Heat Stress” are two separate experimental work which has not been correlated well by the authors. Please improve the manuscript for the coherence of the experimental work and presented results.

1.      The abstract should include the heat stress conditions in which the experiments were performed on the Lycium barbarum plants.

2.      How it can be concluded that the fruits collected at five different stages were actually harvested from stress-free and disease-free plants? Any concurrent abiotic or biotic stress may affect the gene expressions.

3.      Moreover, the details of the total number of biological and technical replicates are not clear.

4.      “The greatest numbers of AQP isoforms are present in plants versus animals and microorganisms, implying their indispensable role in plant sessile living and environmental response [10,11]. The increase in AQP number during the evolution of plants from aquatic to terrestrial and from lower to higher further highlighted the importance of AQPs in the adaptation of higher plants to land life [5,12].” Delete these sentences or shift to the earlier introductory paragraph.

5.      In the introduction section, the perspective of heat stress on Lycium barbarum, impact, and associated morpho-molecular responses are missing. 

6.      Details of the cultivars may be elaborated.

7.      Discussion must be strengthened from a heat stress point of view.

8.      Please clarify whether it was a neighbor-joining method or a maximum likelihood-based phylogenetic tree. 

Author Response

Thank you for your comments. We have provided a point-by-point response to your comments as follows (Cover letter).  In addtion,  we upload the revised manuscript (Manuscript-Response) for your review. Please see the attachment.

Cover letter-Response to Reviewer 2 Comments

Point 1: The abstract should include the heat stress conditions in which the experiments were performed on the Lycium barbarum plants.

Response 1: We have supplemented related information in the corresponding position of the introduction, the current description is as “ Most of their transcripts presented a quick and sharp upregulation response to heat stress following exposure of the 2-month-old seedlings to a 42 ℃ temperature for 0, 1, 3, 12, or 24 h “.

Point 2: How it can be concluded that the fruits collected at five different stages were actually harvested from stress-free and disease-free plants? Any concurrent abiotic or biotic stress may affect the gene expressions.

Response 2: We have added the following description to the corresponding position of 2.1.Plant Materials: Within 30 days before sampling, the mean high temperature and low temperature of Yinchuan were 27.50 °C and 12.37 °C, respectively, while the field were regularly irrigated to keep soil moisture suitable. In addition to rational field management, healthy fruits in good condition will be selected after sampling and used for experiments. 

Point 3: Moreover, the details of the total number of biological and technical replicates are not clear.

Response 3: We have added the related details to ”2. Materials and Methods”.

Point 4: “The greatest numbers of AQP isoforms are present in plants versus animals and microorganisms, implying their indispensable role in plant sessile living and environmental response [10,11]. The increase in AQP number during the evolution of plants from aquatic to terrestrial and from lower to higher further highlighted the importance of AQPs in the adaptation of higher plants to land life [5,12].” Delete these sentence s or shift to the earlier introductory paragraph.

Response 4: We have shift these sentences to the earlier introductory paragraph.

Point 5: In the introduction section, the perspective of heat stress on Lycium barbarum, impact, and associated morpho-molecular responses are missing.

Response 5: Relevant content has been added to the introduction section.

Point 6: Details of the cultivars may be elaborated.

Response 6: We have added the related details to “2.1. Plant Materials”.

Point 7: Discussion must be strengthened from a heat stress point of view.

Response 7: We had modified our manuscript and marked the ralated discussion of the cultivars up using the “Track  Changes” function.

Point 8: Please clarify whether it was a neighbor-joining method or a maximum likelihood-based phylogenetic tree.  

Response 8: The manuscript has been adjusted as follows:The phylogenetic tree was constructed by MEGAX software from a ClustalX alignment of related amino acid sequences (bootstrap replicates = 1000) using the maximum likelihood method.

Round 2

Reviewer 2 Report

The manuscript has been substantially improved.